# OpenReview forum: "Rethinking Sparse Autoencoders: Select-and-Project for Fairness and Control from Encoder Features Alone"
_ICLR.cc/2026/Conference — ICLR 2026 Conference Withdrawn Submission_

### Official Review · Reviewer_TVgc · 2025-10-22

**Soundness:** 2
**Presentation:** 2
**Contribution:** 3
**Rating:** 4
**Confidence:** 4

**Summary:**

This paper introduces the **S&P Top-K method**, which selects salient features from Sparse Autoencoders and applies orthogonal projection in the embedding space to achieve effective debiasing and behavioral control. The approach emphasizes encoder-centric representation steering and contrasts with decoder-based reconstruction methods. The authors conduct extensive experiments across language and multimodal models, demonstrating strong performance in mitigating bias and controlling behaviors such as sycophancy, outperforming established baselines like masked reconstruction.

**Strengths:**

This paper has several strengths:

1. **Cross-model validation**: The method is evaluated on both language and multimodal models, demonstrating strong generality and robustness.
2. **Encoder-centric paradigm**: It shifts focus from decoder reconstruction to encoder feature control, offering a novel and interpretable perspective.
3. **Low integration cost**: Once SAEs are trained, the method requires no additional fine-tuning, making it efficient and practical for downstream use.

**Weaknesses:**

1. **Methodological clarity**

I find several aspects of the proposed method are insufficiently explained. For example, Section 3 lacks sufficient explanation of key steps, such as how the learned weight vector functions as a feature-importance signal, by cosine similarity or other criterion? Also, C mentions “distributional variations across sensitive attribute values” without clearly defining which distribution is being analyzed.

2. **Related work comparison**

The paper is not well-situated in the context of existing studies on SAE-based feature manipulation. Recent works such as

[1] Do I know this Entity? Knowledge Awareness and Hallucinations in Language Models, 2024

[2] Scaling and Evaluating Sparse Autoencoders, 2024

also use SAE-derived features to control model behavior by subtracting decoder-weight features from the representation space. The lack of direct comparison with these methods weakens the empirical positioning.

Moreover, the orthogonal projection technique in S&P Top-K appears conceptually similar to the projection approach used in [1], which should be acknowledged and contrasted more explicitly.

3. **Formatting and grammatical issues**

The paper contains several grammatical errors (e.g., line 113), formatting inconsistencies (line 236), and incomplete references (line 236), which detract from its overall polish.

4. **Limited scope of evaluation**

The experimental coverage could be expanded to include additional model architectures such as Gemma-based SAEs and a broader set of topics related to model trustworthiness, which would strengthen the paper’s generality and impact.

**Questions:**

How would the S&P Top-K method perform if applied to decoder features, and what underlying factors intuitively differentiate encoder features from decoder features in this context?

---

### Official Review · Reviewer_6tCC · 2025-10-30

**Soundness:** 2
**Presentation:** 2
**Contribution:** 3
**Rating:** 6
**Confidence:** 4

**Summary:**

This work proposes improving the inference of sparse autoencoders (SAEs) by projecting the input onto the selected encoder directions, rather than reconstructing it with the decoder. The decoder is only used for training the SAE. Experiments with the CLIP ViT-B/16 and Llama 3 8B Instruct models demonstrate the validity of the approach for steering, specifically debiasing, at inference time.

**Strengths:**

1. This work proposes a significant paradigm shift in controlling large models with SAEs, which has the potential to spark impactful future work in this new direction.
2. Applying the method to two domains (vision, language) demonstrates its versatility and simplicity.
3. The paper is generally well-written and reports results appropriately.

**Weaknesses:**

1. The method lacks theoretical grounding. It seems that some of the steps outlined in Section 3 are trial-and-error approaches without a clear intuition for why approach A would be more appropriate than B, or whether a particular heuristic is even necessary.
2. Experiments are weak:
    - a) The experimental setup is limited to a single SAE architecture and expansion rate $k$ per model.
    - b) The baseline SAE / Masked Reconstruction can be arbitrarily weak without calibrating the steering strength - see [a] and references given there.
    - c) Some experimental results seem inconclusive, e.g. the methods' rankings differ between Table 2 (CelebA) and Table 3 (FairFace).
    - d) The comparison with CAV-like methods (Table 5) should be given in the main text and extended to include its more recent extensions like [b].
3. In [c], SAEs are also used to debias CLIP on CelebA (Figure 6).
4. The paper should include code to reproduce the results.

[a] To steer or not to steer? Mechanistic error reduction with abstention for language models. ICML 2025

[b] Navigating neural space: Revisiting concept activation vectors to overcome directional divergence. ICLR 2025

[c] Interpreting CLIP with hierarchical sparse autoencoders. ICML 2025

**Questions:**

1. What does '(re)training-free' mean in the context of this work? It is used only twice: in the abstract and in contribution 2.
2. More details are needed regarding the "S&P Top-K + BendVLM" method in the paper, beyond what's written in L273.
3. How can results for the Orth-Proj and Orth-Cal methods be sourced from (Gerych et al., 2024)? Is the experimental setup and implementation identical?

**Other feedback**:
- The term 'Masked Reconstruction' is confusing. It cannot be found in the papers referenced next to it in L166 (Cywinski & Deja, 2025; Anthropic, 2024), and it is not conventionally used in the context of SAEs. This appears to be a new term, which makes the Abstract and Introduction unclear, particularly Figure 1.
- Figure 1 conveys little information. It's a single example where method A leads to a better prediction than method B. On another example, it might be the opposite. Why did it happen? How do methods A and B differ? How general is this example?
- Figure 2 is not understandable without actually reading the paper. What does interpolation do? What is the single bias axis? It overlooks the fact that the goal of the method is to remove information about the concept (orthogonalized against). Additionally, placing two figures before the Introduction is excessive.
- The Stylist approach proposed by Smeu et al. (2025) is a major factor contributing to the method's performance, so it should be briefly described in the main text or the Appendix.
- L145: typo in $z$ should be `\bm{z}`
- L158: what is $\mathbf{Y}$?
- L236: typo in the citation "?"
- L315: "Worst Group AUC ROC" should be described in the paper
- Many typos in how quotation marks are written.

---

### Official Review · Reviewer_Xg7k · 2025-10-31

**Soundness:** 3
**Presentation:** 2
**Contribution:** 2
**Rating:** 2
**Confidence:** 4

**Summary:**

This paper introduces S&P Top-K, an encoder-centric method for model steering (e.g. controlling model behavior). It selects the Top-K SAE encoder features (instead of a sum of the decoder weights) that are most related to a target attribute and can combine them into a single control direction. At test time, the method removes the target information by projecting the model’s embedding away from that direction.
It requires no retraining and adds very little computation. The approach is evaluated on vision-language fairness tasks and LLM behavior steering, and they report consistent improvements over traditional methods.

**Strengths:**

- New viewpoints: this paper proposes a different view that shift the focus from decoder-centric to encoder-centric SAE interventions.
- Computationally efficient and plug-and-play: no model retraining and only a single projection operation at inference.
- Good performance: experiments on both VLM and LLM shows effectiveness of the method.

**Weaknesses:**

- Motivation & problem framing are weak. The paper does not convincingly justify why encoder-centric intervention is necessary beyond empirical gains. It does not clearly articulate why an encoder-centric intervention is necessary or what specific failure modes of existing decoder-based SAE approaches motivate the new design.
Even if the paper aims to “rethink” the intervention mechanism, it lacks a modeling or analytical treatment of existing methods, which further limits the clarity and strength of its claimed novelty.

- Lack of related works
The related-work section does not adequately cover prior studies, e.g.
[1] SEMANTICS-ADAPTIVE ACTIVATION INTERVENTION FOR LLMS VIA DYNAMIC STEERING VECTORS, ICLR'2025
[2] INTERPRETABLE STEERING OF LARGE LANGUAGE MODELS WITH FEATURE GUIDED ACTIVATION ADDITIONS, ICLR'2025 workshop
[3] IMPROVING INSTRUCTION-FOLLOWING IN LANGUAGE MODELS THROUGH ACTIVATION STEERING, ICLR'2025
[4] Analyzing the Generalization and Reliability of Steering Vectors, NIPS'2024
[5] Do I Know This Entity? Knowledge Awareness and Hallucinations in Language Models, NIPS'2025
[6] Evaluating feature steering: A case study in mitigating social biases, Anthropic
...

- Lack of theoretical grounding.
The method is described procedurally, but the theory behind the encoder-centric is not presented. It also lacks analysis of identifiability, feature entanglement, or projection optimality, leaving the approach without a solid conceptual foundation.

- Limited or unfair Evaluation
Since the paper discusses a general methodological paradigm, choosing only fairness scenarios for comparison is not comprehensive enough. Other aspects, such as trustworthiness, knowledge entities, social bias, etc., should also be considered. Also, comparison with related methods is necessary and required to be discussed.

- Experiments concerns.
1. Key sensitivities (SAE training data/quality, choice of k, α, and whac-a-mole effects) are not thoroughly explored;
2. The time cost and the efficiency should be discussed;
3. Behavioral evaluation on LLMs relies heavily on an internal “LLM-as-a-judge” protocol (potentially subjective and model-dependent) with limited human evaluation and limited reporting of statistical significance/variance.

**Questions:**

see weakness.

---

### Official Review · Reviewer_pdw3 · 2025-11-01

**Soundness:** 2
**Presentation:** 2
**Contribution:** 2
**Rating:** 2
**Confidence:** 2

**Summary:**

This paper revisits sparse autoencoders and proposes an encoder-centric framework for model steering and debiasing without relying on the decoder. Traditional SAE-based steering methods modify sparse latent activations and decode them back into the model’s embedding space, which can introduce reconstruction errors and high computational cost.

The proposed method instead directly operates on encoder representations: it selects the top-K encoder features correlated with a target attribute or behavior, optionally aggregates them into a single control axis, and applies an orthogonal projection in the embedding space to suppress or enhance specific semantic directions.

The framework is training-free and lightweight, demonstrated on vision–language models (CelebA, FairFace) for fairness debiasing and large language models (Llama-3-8B) for behavior control (reducing aggressiveness and sycophancy), showing up to 3.2× and 3.6× improvements over decoder-based masked reconstruction baselines.

**Strengths:**

The paper's primary originality lies in its proposal to rethink the standard application of Sparse Autoencoders for model steering. It shifts the paradigm from a "decoder-centric" approach to an "encoder-centric" one. This work identifies and challenges the implicit assumption that SAE semantics are primarily stored in the decoder.

A key finding is that while traditional masked reconstruction significantly harms downstream task utility (wgAUC), the proposed S&P Top-K method (with interpolation) successfully preserves it .In LLM experiments, the method demonstrates substantial gains over the baseline (e.g., 3.6x improvement in sycophancy reduction)

By identifying a key weakness in the conventional "masked reconstruction" approach—namely, poor utility preservation and ineffective control —the paper offers a practical and more effective alternative. The proposed S&P Top-K method is computationally lightweight, training-free, and demonstrates superior performance. These findings provide a valuable contribution that could inform future "best practices" for applying SAEs to model steering and debiasing.

**Weaknesses:**

Limited methodological novelty: The core contribution mainly reuses existing components of sparse autoencoders and linear subspace projection. The idea of encoder-only steering through orthogonal projection has conceptual overlap with prior works. To enhance originality, the authors could integrate a theoretical analysis of how sparsity contributes to the interpretability or robustness of the learned control directions, rather than merely showing empirical improvements.

Insufficient interpretability and visualization: While the paper argues that sparsity leads to interpretable feature directions, it does not provide qualitative visualizations (e.g., activation heatmaps or t-SNE plots of projected embeddings). Without showing how selected Top-K features correspond to semantic concepts, it is difficult to validate interpretability claims. Providing such visualization would strongly reinforce the motivation of sparsity-driven controllability.

Algorithms are limited: Its success depends on a complex and carefully tuned process, where the interpolation method must be LP weighted and feature selectors must be trained for the data. The high degree of process dependency and extreme sensitivity to specific component selection greatly increase the difficulty of method tuning and weaken its usability

**Questions:**

Provide theoretical explanations. Why does LP bring stability while Mean/Sign fails;

Can we visualize the features before and after modification to prove that certain regions have biased features?

The model results may be sensitive to the value of K and the random selection process. It is recommended to provide performance curves for different K values.

The article mentions “For each targeted behavior (aggressiveness and sycophancy), we train a logistic regression classifier on the SAE activations to distinguish between the behavior and its corresponding non-behavior counterpart (i.e., aggressive versus non-aggressive, and sycophantic versus non-sycophantic responses).” does it need to train a feature selector for each requirement

---

### Note · Authors · 2025-11-19

I have read and agree with the venue's withdrawal policy on behalf of myself and my co-authors.